# Media Reports about Violence against Medical Care Providers in China

**DOI:** 10.3390/ijerph18062922

**Published:** 2021-03-12

**Authors:** Liheng Tan, Shujuan Yuan, Peixia Cheng, Peishan Ning, Yuyan Gao, Wangxin Xiao, David C. Schwebel, Guoqing Hu

**Affiliations:** 1Department of Epidemiology and Health Statistics, Hunan Provincial Key Laboratory of Clinical Epidemiology, Xiangya School of Public Health, Central South University, Changsha 410078, China; hersey.tan@foxmail.com (L.T.); yuanshujuan@csu.edu.cn (S.Y.); chengpeixia@csu.edu.cn (P.C.); ningpeishan@csu.edu.cn (P.N.); gaoyuyangyy@csu.edu.cn (Y.G.); xiaowangxin@csu.edu.cn (W.X.); 2Department of Psychology, University of Alabama at Birmingham, Birmingham, AL 35294, USA; schwebel@uab.edu; 3National Clinical Research Center for Geriatric Disorders, Central South University, Changsha 410078, China

**Keywords:** China, doctors, media reports, physicians, violence against medical care providers

## Abstract

Improper, unprofessional, or misleading media reports about violence against medical care providers (typically doctors and nurses) may provoke copycat incidents. To examine whether media reports about violence against medical care providers in China follow professional journalism recommendations, we identified 10 influential incidents of violence against medical care providers in China through a systematic strategy and used standardized internet-based search techniques to retrieve media reports about these events from 2007–2017. Reports were evaluated independently by trained coders to assess adherence to professional journalism recommendations using a 14-item checklist. In total, 788 eligible media reports were considered. Of those, 50.5% and 47.3%, respectively, failed to mention the real and complete names of the writer and editor. Reports improperly mentioned specific details about the time, place, methods, and perpetrators of violence in 42.1%, 36.4%, 45.4%, and 54.6% of cases, respectively. Over 80% of reports excluded a suggestion to seek help from professional agencies or mediation by a third party and only 3.8% of reports mentioned the perspectives of all three key informants about an event: medical care providers, patients, and hospital administrators. Of those that mentioned medical care providers, patient, and/or hospital administrator perspectives, less than 20% indicated they had obtained the interviewee’s consent to include their perspective. We concluded that most reports about violence against medical care providers in the Chinese media failed to strictly follow reporting recommendations from authoritative media bodies. Efforts are recommended to improve adherence to professional guidelines in media reports about violence against medical care providers in China, as adherence to those guidelines is likely to reduce future violent events against medical care providers like doctors and nurses.

## 1. Introduction

Violence against medical care providers, formally labelled as “workplace violence in the health sector” by the World Health Organization (WHO), refers to incidents when health workers are abused, threatened, or assaulted in circumstances related to their work, including commuting to and from work, and involving an explicit or implicit challenge to their safety, well-being, or health. It includes both physical and psychological violence [1].

Violence against medical care providers like doctors and nurses is a surprisingly common event that significantly threatens the safety of health care workers in many countries. A 2007 survey involving 3465 nurses from the US Emergency Nurses Association reported that approximately 25% and 20% of respondents experienced physical violence and verbal abuse, respectively, in the previous three years [2]. Similarly, a survey of 16,327 medical practitioners in Australia found that 71% of them reported experiencing verbal or written aggression and 32% physical aggression [3]. The annual UK National Health Service (NHS) staff survey in 2019 revealed that 15% of NHS members reported experiencing at least one incident of physical violence from patients, their relatives, or other members of the public over the previous 12 months [4].

Violence against medical care providers is also common in Asia. A survey of medical ward staff in India found that 57% experienced violence from patients or visitors [5]. In China, the number of recorded violent events against doctors increased from 9831 in 2005 to 17,243 in 2010 [6]. According to a national survey of 316 Chinese hospitals, the proportion of medical staff who experienced verbal or physical violence from patients increased from 90% to 96% between 2008 and 2012 [7]. Chinese Medical Doctor Association data showed that 60% of health workers suffered verbal abuse and 13% were physically attacked in 2014 [8].

Violence against medical care providers brings various adverse consequences to health care workers, including increased psychological stress, increased employee turnover, decreased job satisfaction, decreased productivity, and decreased trust in management and colleagues [9,10]. Beyond consequences for health care workers themselves, violence against medical care providers in health care facilities may also indicate widespread tensions between medical care providers and their patients, which may compromise the accessibility and quality of the health system [10,11].

A range of factors is related to the likelihood of incidents of violence against medical care providers. Factors documented in previous work include long waiting time for appointments [12], unrealistic expectations for health cures from patients (e.g., complete recovery after one or two visits) [13], limited legal channels for resolving medical disputes [14,15], and absence of legal penalties against violence [14,16].

Previous studies suggest media reports that inappropriately detail conflicts between medical care providers and patients [17] or that fail to present the perspectives of the doctors and hospital administrators [18] may worsen tense doctor–patient relationships and could produce copycat or repeated incidents of violence. Such an impact is likely magnified in recent years due to the flourishing development of social media and its power to spread news stories broadly and quickly [19].

Only a few previous publications assess media reports about violence against medical care providers from a social science or public health perspective. Hasan and colleagues used quantitative strategies to assess 56 textual news stories in Bangladesh and found violence against medical care providers in 54 (96%) of the 56 stories involved physical violence [20]. Two graduate theses in China used qualitative methods to analyze news reports. Xie analyzed 276 media reports and 198 video stories and concluded that violent events against medical care providers were not reported objectively by the media [21]. Xiong examined 68 media reports of 10 events of violence against medical care providers occurring between 2009 and 2012 and concluded that the media did not report the events accurately or comprehensively [22]. Although valuable to the field, these studies did not detail the problems present in media reports and, thus, offer only a first step toward implementation of achievable solutions to improve media reports.

This study used systematic quantitative strategies to assess whether media reports about violence against medical care providers follow recommendations from authoritative journalism societies and associations that publish guidelines on how journalists should report news; whether the media stories include the perspectives of patients, medical care providers, and hospital administrators; and whether the consent of interviewees was obtained when interviewees’ views were included.

## 2. Materials and Methods

Our research proceeded in four steps: (1) identifying influential violent events against medical care providers for inclusion in the research study; (2) developing search terms and criteria for media report inclusion; (3) searching and screening media reports about the influential violent events to include in the detailed study; and (4) developing evaluation checklist and evaluating the media reports.

### 2.1. Step 1: Identifying Influential Violent Events against Doctors for Inclusion

Because there were a large number of potentially relevant media reports about violence against medical care providers, we limited our research to reports for the most important and influential events. We implemented a systematic process to identify such important events for inclusion in our study. To start, we consulted the annual “hot event” list published by the People’s Daily Online Public Opinion Monitoring Room [23,24,25,26,27,28,29,30,31,32,33]. This list is developed by the national media agency, the People’s Daily, based on internet-based public opinion about the most important hot events that attract online readership in China [34]. Currently, the hot event list is the only domestic source ranking the social influence of news events. It is released annually.

The People’s Daily annual hot event lists between 2007 and 2017 included only one incident of violence against medical care providers, “Maternal death in Xiangtan”, which we, therefore, used as a seed to search for related events using the Baidu News search engine. Baidu News is the largest and most up-to-date online search engine in China; it collects and links to news stories from over 500 websites [35]. Baidu News offers a search feature to suggest links of about 5–6 reports relevant or related to the one the user reads [36]. With the People’s Daily hot event story as a seed and entered it into the Baidu News search engine, we discovered several related and relevant stories. Those stories were then entered as seed events for the next round of searching. The process continued in an iterative manner until no additional events of violence against medical care providers were identified.

Criteria for events discovered through this seeding process and eligible for inclusion in our study were threefold: (1) occurred between January 2007 and December 2017 and reported in an online Chinese media source; (2) reporting about an event that occurred in a Chinese hospital or other medical institution or in a situation related to medical work (e.g., while commuting to/from work) and involving violence between patients or their family members and medical professionals (e.g., doctors or nurses); and (3) stories reporting an event involving physical or verbal violence against medical care providers. The systematic process of iterative searching identified 10 influential events.

### 2.2. Step 2: Developing Search Terms and Criteria

We adopted a strategy from previous research to develop 76 combinations of search terms concerning each of the 10 included events [37]. Terms included three categories (place of event, individuals involved, event abbreviations in Chinese) for each event. Again seeking to focus our inclusion criteria on the most important, influential, and impactful events, and recognizing the volume of online media sources available in China, we limited searches to news websites with large or moderate readership and impact. Eligible news websites were those with an Alexa ranking of 20,000 or less and a PageRank of 7 or higher. Alexa rankings are a global website-ranking system based on multiple evaluation indexes including comprehensive ranking, visits ranking, and page visits ranking [38]. Lower scores reflect websites with greater impact and readership. PageRank measures public interest and attention to websites; a PageRank greater than 7 indicates that the website is “very popular” [39]. Together, Alexa rankings and PageRank are widely recognized as valid indicators of a website’s social influence and impact. Websites published in languages other than Chinese, inaccessible to the public in mainland China, and those not designed to report news to the public were also excluded.

Among the news websites included, we adopted previous strategies to divide the websites into three categories: portal news website, national news website, and local news websites [40]. Portal news websites are those owned by commercial companies rather than government agencies. National and local news websites are administered by the central (national) and local governments, respectively. National news websites in China tend to disseminate news of nationwide interest and influence whereas local news websites focus on stories of interest and influence both nationally and locally.

### 2.3. Step 3: Searching and Screening Media Reports

Using the selected search strategies, we performed searches for media reports on the Advanced Baidu News platform. Since most media reports for violence against medical care providers from over 10 years ago were not available online when our research was conducted, we limited the years of our search for reports to 2007–2017. Python programming was developed to search and automatically save the searched reports into a MySQL database. The MySQL database included the full-text media reports plus six objective fields for each media report: an assigned ID number, the title of the media report, the website link to the original report, the date of publication, the name of the source website, and the search terms used to identify the report. The search was conducted between August and October 2018. Preliminary searches captured 17,221 media reports after excluding duplicate reports based on an identical URL, including 6030 from portal news websites, 9850 from national news websites, and 1341 from local news websites (Figure 1). After the database was created, we used two steps to screen for eligible reports. First, we manually reviewed report titles and removed reports that did not include the relevant search terms or that consisted of incorrect search terms in the title. (Note: Unlike academic datasets such as PubMed, Baidu News does not support precise searches and therefore can generate irrelevant records in the search [38]). This step excluded 13,388 irrelevant reports (Figure 1). Second, we reviewed the full-text reports remaining after the first step and excluded reports based on the following criteria: (1) not involving violence against medical care providers; (2) irrelevant to the 10 selected events; or (3) not expected to include aspects of the evaluation checklist below, such as editorial opinion pieces rather than news stories. The second step removed an additional 3045 ineligible reports.

This process left 788 eligible media reports concerning six influential events about violence against medical care providers: 465 from portal news websites, 315 from national news websites, and eight from local news websites. No eligible reports were collected for the other four influential violent events against medical care providers.

### 2.4. Step 4: Developing Evaluation Checklist and Evaluating the Reports

The research group, which consisted of five doctoral-level public health faculty members and five public health graduate students, considered China’s code of professional ethics for journalists [41] and three sets of relevant international guidelines [42,43,44] to create a list of objective criteria to be used to evaluate the extent to which each media report met international and national standards for proper media reporting. Each individual guideline in each of the four broad sets of guidelines was discussed in turn to assess the technical and practical feasibility of transforming the recommendation into an operable and objectively coded item that could be used by the research team. Following two rounds of extensive group discussions, the research group selected initial items for inclusion from the four considered guidelines. Those items comprised the initial draft of the evaluation checklist. The draft checklist was pilot tested and refined through evaluation of 50 media reports. Last, a final checklist of 14 items was created. The final checklist included items that were grouped into three parts: adherence to professional recommendations (eight items); balanced inclusion of the perspectives of patients, medical care providers, and hospital administrators, which reduces bias in reporting and provides all perspectives on the event (three items); and evidence that interviewees provided consent to include their perspectives (three items). All checklist items could be scored in an objective, yes/no manner. The complete checklist appears in Table 1.

Evaluation of the 788 eligible reports was completed by trained raters using the 14-item checklist. Following that review, an independent and experienced coder (the first author, L.T.) re-evaluated all reports to detect and correct any errors. EpiData 3.1 software was used to enter the data.

### 2.5. Data Analysis

The number and percentage of media reports about violence against medical care providers adhering to the recommendations were calculated and graphed descriptively in bar charts.

### 2.6. Ethical Approval

This analysis was approved by the ethics committee of Xiangya School of Public Health, Central South University (No. XYGW-2018-24).

## 3. Results

### 3.1. Adherence to Professional Recommendations

Figure 2 presents the percentage of media reports not adhering to the professional recommendations among the respective sources of portal, national, and local news websites. Of the 788 media reports, 398 (50.5%) and 373 (47.3%) reports did not use the real and complete names of the writer and editor. Across the three types of websites, portal news websites had the highest percentage of failing to include the name of the writer (53.3%) and national news websites had the highest percentage of failing to include the name of the editor (61.3%).

About 40% of the 788 reports inappropriately included details concerning the violent events against medical care providers (42.1% included the time of the violence, 36.4% the place, 45.4% the methods, and 54.6% the characteristics of the perpetrator(s)). Strikingly, over 80% of 788 media reports did not include information for the public on how to seek help from professional agencies or how to seek mediation from a third party when they feel they were treated unfairly as patients.

### 3.2. Balanced Presentation of Medical Care Providers, Patient, and Hospital Administrator Perspectives

Of the 788 media reports, only 3.8% mentioned the perspectives of all three relevant parties (Table 2). Reports that mentioned the perspectives of just two of the three parties accounted for very small proportions of the 788 reports––5.5% for patient and medical care providers, 6.7% for patient and hospital administrator, and 4.2% for medical care providers and hospital administrator. The remaining reports included 14.5%, 3.4%, and 3.7% that mentioned the perspective of one party––the patient, medical care provider, and hospital administrator, respectively––and 58.2% that failed to mention the perspective of any of the three parties. Substantial differences existed across the three types of websites for presenting views of the three parties.

### 3.3. Interviewee Consent

Among the media reports that incorporated patient perspectives, 11.2% (27/240) indicated gaining the consent of the patient. Comparable proportions were 15.8% (21/133) for reports mentioning medical care providers’ perspectives and 18.6% (27/145) for reports mentioning hospital administrators’ perspectives (Table 3). The proportion varied substantially across the three types of websites.

## 4. Discussion

### 4.1. Key Findings

This study used quantitative methods to evaluate 788 media reports concerning influential violent events against medical care providers in China. We generated three key findings. First, half or more of the included reports did not follow each of the professional journalistic recommendations for proper reporting (mention real and complete names of the article’s writer and editor, properly present details of the violent events with specific details omitted, include information for readers on how to seek help from professional agencies or mediation by a third party in case of ideation about committing violence against medical care providers). Second, most reports failed to balance reporting concerning the perspectives of patients, medical care providers, and hospital administrators. Third, when patient, medical care provider, and hospital administrator perspectives were obtained, very few reports indicated they had obtained consent from interviewees to include their perspectives.

### 4.2. Interpretation of Findings

According to both Chinese and global professional journalism standards [41,42,43,44], it is recommended to include the real name of writers and editors in news reports. Only about half the reports concerning violence against medical care providers in our study complied with this recommendation. One possible reason for this result is that many emerging news websites and platforms fail to hire journalists and editors who have professional education or training on media reporting [45]. For example, many news websites use batch processing and crawler programs to automatically obtain news from other media sources and release the stories on their own websites [46]. A second likely contributing factor is the lack of internal supervision within the website platforms or external supervision from the government. Despite some recent initial progress, Chinese governmental supervision to ensure media reports follow the professional recommendations remains at an early stage [47]. There is no published evidence to suggest internal supervision occurs at most Chinese media outlets.

Inclusion of explicitly risky or violent details in popular media sources without taking the precautions recommended by international journalism standards may increase societal interest in violent and dangerous behaviors, arousing interest and possible imitation through social learning and modeling [48,49]. Despite this, approximately 40% of the media reports we reviewed inappropriately included explicit details concerning violence against medical care providers. Websites may purposely include this information to attract readers. In the face of fierce competition for the journalism market in China, all media sources struggle to attract audiences and increase their social influence [50], a goal that other researchers have suggested may be boosted by offering details about violent events like suicide [51]. Similar to suicide events, violence against medical providers are socially sensitive events that can involve violent details. International journalism standards recommend that reporters avoid describing the details of such events in media reports.

It is also recommended by professional journalism standards that media reports about violence, suicide, drug overdoses, and similar situations include information for readers on how to seek expert advice for prevention [52,53]. There is evidence, for example, that prevention-oriented media reports are associated with subsequent decreases in suicide within a population [54]. However, over 80% of media reports in our study excluded such information. Our finding is not unique. An analysis of 444 Australian media reports about violence against women showed a similar result. For example, over 95% of reports excluded information on victim or perpetrator services (e.g., helplines, advocacy, or counseling services) [55]. One possible explanation for this finding is that Chinese journalists may not have requisite knowledge and awareness to incorporate appropriate information in their stories. Most domestic journalists in China receive education in journalism and communication, training that typically excludes learning about inclusion of help-seeking information in media reports [41].

High-quality journalism incorporates the perspectives of all sides, which suggests that media reports about violence against medical care providers should incorporate at least three perspectives: those of patients, medical care providers, and hospital administrators [56]. However, our results suggest patients’ perspectives were much more likely to be reported compared to those of medical care providers and hospital administrators (14.5% vs. 3.4% and 3.7%), and all three perspectives were omitted from a large portion of the media reports. When those perspectives were shared, over 80% of reports did not specify whether consent of the interviewee was obtained, an ethical prerequisite for sharing views in media reports [41,42,43,44].

Consent may have been obtained but not mentioned in the stories in some cases. Similarly, perspectives may have been sought but not offered. For those reports that included the interviewees’ perspectives but did not actually obtain the consent of interviewees, lack of professional education and training among journalists and the fierce competition between media companies are likely factors that led to such results. Journalists may prefer to report a breaking news story quickly in an attempt to attract the public’ attention, even if they fail to collect appropriate perspectives in a proper way (namely, informing the interviewees and obtaining their consent before gathering the perspectives of interviewees). For example, the “Maternal death in Xiangtan” event involved the death of a pregnant woman from amniotic fluid embolism in Hunan Xiangtan County Maternity and Child Health Hospital on 10 August 2014 and led the decedent’s family members to multiple acts of violence, including physical attacks on doctors and nurses, forcing medical staff to kneel in front of the decedent’s body, and blocking of the hospital gate. In early stages of media reporting about the event, statements from the patient’s family members were prominently included, but the perspectives of the medical care providers and hospital administrators were omitted. As a result, descriptive words like “bloody”, “nude,” “tragic death,” “absence of medical staff in the operating room,” and “an unidentified man was found smoking in the operating room” emerged in media reports, exaggerating details of the event and creating inappropriate critiques and attacks toward the medical care providers and hospital [57]. In fact, the final medical investigation by the local health department found the death was caused by multi-organ failure that was initiated by amniotic fluid embolism. The medical staff had not committed any errors during the process of diagnosing and treating the patient [58].

### 4.3. Policy Implications

Our findings have implications to highlight the importance and urgency of improving how events of violence against medical care providers are reported by the media in China. We recommend three specific policy actions. First, either the Chinese government or a prominent professional journalism association in China should improve education and training for journalists to raise their competency and awareness. Strict and standard qualification exams should be implemented nationwide for all journalist candidates (including writers and editors), and the training and exams should incorporate training on professional standards concerning the reporting of violent events. Second, internal reviews should be conducted for all media reports, including both traditional and online media, to ensure they meet professional standards. Third, central and local governments should strictly and regularly monitor media reports about violence against medical care providers to assure their adherence to professional reporting standards, thus maximizing the role of media in creating a safe and positive relationship between medical care providers and their patients and preventing unwanted violence against medical care providers. Automated monitoring systems might be developed by the government to detect media reports out of compliance with professional recommendations, facilitating the government oversight process. Without such steps, continued media reports that violate international standards could create continued risk of harm to medical providers in China.

### 4.4. Study Limitations

This study has three major limitations. First, we limited our search to 10 influential events involving violence against medical care providers. Other events occurred during the study time period in China, but including additional events would be unlikely to change our findings, as all domestic journalists receive similar education and work under the same government regulations [41,59]. Second, our checklist was limited to data that we could evaluate reliably and feasibly. We omitted other professional recommendations, such as adherence to objective reporting, because we were unable to verify whether those recommendations were followed or not. Third, we only focused on Chinese media reports about violence against medical care providers. Our findings may not generalize to other countries and cultures.

## 5. Conclusions

A harmonious relationship between medical care providers and patients is vital to the development of a well-functioning medical care system. In an era of explosive information that is ubiquitously available, media sources play an increasingly central role in how citizens understand and relate to their medical providers. Media also can influence the public’s perception of unwanted events like violence against medical care providers. Unfortunately, the majority of domestic media reports about violence against medical care providers in China did not comply with professional journalism recommendations. They frequently failed to list the real and complete names of writers and editors, improperly revealed details of the violent event, and excluded mention of prevention strategies or suggestions for resolving conflicts. They also reported the perspectives of medical care providers, patients, and hospital administrators in an imbalanced manner and failed to mention interviewee consent to share those perspectives when provided. We recommend improved education and training for journalists in China, enhanced internal review within media sources, and increased government monitoring of Chinese media to reduce improper media reports of violence against medical care providers, thus reducing unwanted copycat violence events by readers.

## Figures and Tables

**Figure 1 ijerph-18-02922-f001:**
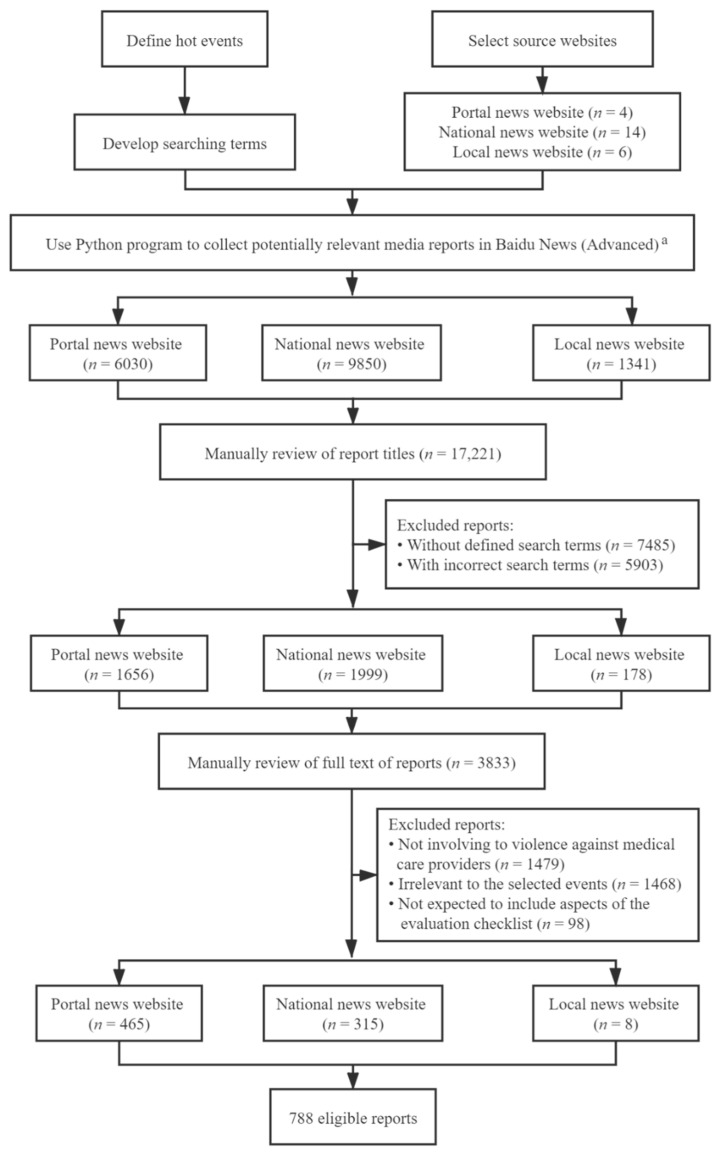
Flow chart of selection of eligible media reports about violence against medical care providers. Note: ^a^ Unlike academic datasets, Baidu News does not support exact searching and therefore can generate irrelevant records.

**Figure 2 ijerph-18-02922-f002:**
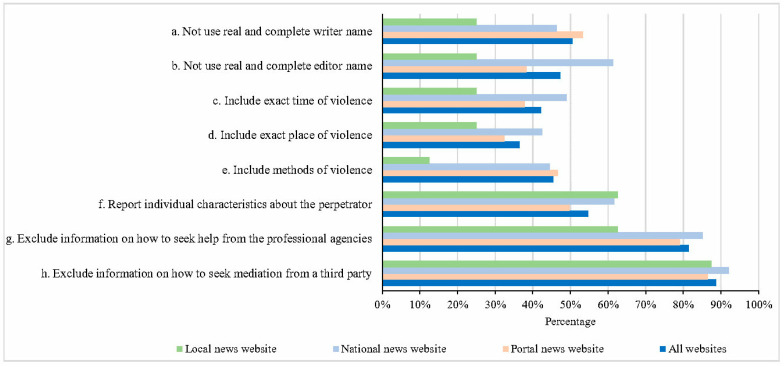
Percentage of media reports about violence against medical care providers adhering to professional recommendations (%).

**Table 1 ijerph-18-02922-t001:** Fourteen-item checklist to evaluate adherence to professional journalism recommendations.

Fourteen-Item Checklist
1. Following professional recommendations for media reports
a. Use real and complete writer name
b. Use real and complete editor name
c. Improperly include exact time of violence
d. Improperly include exact place of violence
e. Improperly include method of violence
f. Improperly report individual characteristics about the perpetrator
g. Include information on how to seek help from professional agencies
h. Include information on how to seek mediation from a third party
2. Balancing perspectives of the patient, medical care providers, and hospital administrator
a. Include perspectives of patient
b. Include perspectives of medical care providers
c. Include perspectives of hospital administrator
3. Obtaining permission from interviewees
a. Indicate consent from medical care providers when their perspectives are included
b. Indicate consent from patients when their perspectives are included
c. Indicate consent from health administrators when their perspectives are included

Note: A yes/no binary option was used to evaluate all 14 items. Item g signifies information on how to seek help from a division within the Chinese health commission to take charge of the management of hospital (including dealing with the patient’s complaints), while item h indicates information on how to seek help from a mediator that is typically outside the health department, such as in a court of law court or a public security office.

**Table 2 ijerph-18-02922-t002:** Percentage of Chinese media reports about violence against medical care providers mentioning the perspectives of medical care providers, patients, and hospital administrators.

Category	Total	Portal News Website	National News Website	Local News Website
a. Only mention the view of patient	14.5%	14.2%	14.6%	25.0%
b. Only mention the view of medical care providers	3.4%	3.7%	3.2%	0.0%
c. Only mention the view of hospital administrator	3.7%	4.7%	2.2%	0.0%
d. Mention the view of patient and medical care providers	5.5%	3.9%	7.6%	12.5%
e. Mention the view of patient and hospital administrator	6.7%	5.4%	8.9%	0.0%
f. Mention the view of medical care providers and hospital administrator	4.2%	3.9%	4.8%	0.0%
g. Mention the view of all three parties	3.8%	4.3%	3.2%	0.0%

Note: In total, 788 valid media reports were included in analysis, including 465 from portal news website, 315 from national news websites, and eight from local news websites.

**Table 3 ijerph-18-02922-t003:** Number and percentage of media reports about violence against medical care providers that included the perspective of patients, medical care providers, or hospital administrators and indicated obtaining consent from those interviewees.

Type of Interviewee/Website	Total Including Perspective	Number Obtaining Consent (Percentage, %)
Patient		
All websites	240	27 (11.2%)
Portal news website	129	21 (16.3%)
National news website	108	6 (5.6%)
Local news website	3	0 (0.0%)
Medical care providers		
All websites	133	21 (15.8%)
Portal news website	73	16 (21.9%)
National news website	59	5 (8.5%)
Local news website	1	0 (0.0%)
Hospital administrator		
All websites	145	27 (18.6%)
Portal news website	85	27 (31.8%)
National news website	60	0 (0.0%)
Local news website	−	−

Note: No reports from local news websites mentioned the perspective of the hospital administrator.

## Data Availability

Aggregate data concerning our analyses are available from the corresponding author, upon reasonable request.

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
