# Peer review of "Media Reports about Violence against Medical Care Providers in China"

_ijerph, 2021, doi:10.3390/ijerph18062922_

Round 1
Reviewer 1 Report
I see the progress of the manuscript, however, it's still difficult for me to agree with with "medical care providers" term. In my humble opinion, and according to the subject literature, violence against nurses and MD differs in it's motivation and consequences.
Reviewer 2 Report
The manuscript presents the results of an investigation that analyzes 788 reports on situations in which any kind of violence has been exercised against medical care providers in China between 2007 and 2017.
The results show that a very significant proportion of these reports fail to comply with different standards of journalistic practice. It is particularly worrying that a significant number of reports provided probably unnecessary details about the specific circumstances in which the events occurred, which may lead to the imitation of these events.
Overall, the article is correctly written.
Here are some specific questions that I think the authors should review:
1. It would be interesting to explain very briefly the reasons for the items on the instrument that may be less obvious: c, d, e and f.
Some readers less familiar with newspaper writing standards may not understand the reasons for these items.
2. Tables 2 and 3.
Although the differences between Portal news websites, National news websites and Local news website can be clearly appreciated through the percentages, it would be interesting if the authors could calculate a statistic that would show whether these differences reach statistical significance or not.
3. Perspectives on the violent event of the patient, the medical care provider and the hospital administrators.
In some cases, it is possible that one of these three agents refused to give the media their point of vew about the event. In these cases, it is obvious that the perspective of that agent cannot be included.
This question should be taken into account when evaluating the result offered in Table 3.
If possible, it would also be interesting for the authors to count in how many news items the media asked the three parties for their vision of the incident, but for some reason one of these parties involved refused to offer information in this regard.
4. Lines 377-383.
This suggestion should be modified or deleted. It is very positive that governments make efforts to monitor compliance with international recommendations on journalistic practice.
However, the disbandment of a media source is an extreme measure that would go against freedom of expression, so it is not a measure that should be taken due to non-compliance with this type of recommendation.
Instead, it would be interesting to increase efforts to improve the training of the media (as indicated in the conclusions); and also of the potential readers of that media, so that they have a greater capacity to identify journalistic practices that skew reality.
The qualification of this idea of ​​the disbandment of a media source is, from my point of view, the most relevant aspect that should be modified in the manuscript.
4. Formal issues.
4.1. Why are decimal numbers written as 50 · 5%, instead of 50.5%?
4.2. Keywords should be in alphabetic order.
Author Response
Please see the attachment

This manuscript is a resubmission of an earlier submission. The following is a list of the peer review reports and author responses from that submission.
Round 1
Reviewer 1 Report
Dear authors. Thank you very much for sharing such valuable data and interesting results.
I consider it necessary to better justify the study: why are these findings important? What impact do they have on health? Who are the primary, secondary and tertiary affected? and what would be the possible effects in the short, medium and long term? The conclusions can also be improved considerably. I suggest including new research questions that arose in the development of this study.
Reviewer 2 Report
Review
Media reports about violence against doctors in China
Liheng Tan , Shujuan Yuan , Peixia Cheng , Peishan Ning , Yuyan Gao , Wangxin Xiao , David C Schwebel , Guoqing Hu
Brief summary of the paper
The manuscript reports the results of a literature search and evaluation of violent events against medical personal in China. Ten events were found and four six of them media reports were analysed (788). A checklist was created and every media report was evaluated whether it adhered to that checklist. Results show that a lot of the media reports did not give the full name of the writer or editor, but gave explicit details about the violent behaviour. The authors seem to believe these media reports to incite further violence against medical staff and conclude that journalists should be educated more and media outlets should be regulated more by the government.
Major aspects:
- Line 69: “Another factor that has been cited as a risk is the presence of misleading or improper media reports that may provoke violence by patients [17].” à This sentence implies that misleading media reports risk violent behaviour of patients against doctors. However, as far as I see it, the WHO report you cite is about violence of patients against themselves (i.e. suicide), how to prevent that and how to prevent copycat behaviour (i.e. more suicides). Therefore, this sentence does not connect to the previous paragraph and is in itself quite misleading.
- Line 70: “Previous studies suggest inappropriate media reports that detail conflicts between doctors and patients, or that fail to present the perspectives of the doctors and hospital administrators, may worsen tense doctor-patient relationships and could produce copycat or repeated incidents of violence [18–20].” à Please be more careful with your citations. The only mention of media in source 20 (He AJ, Qian J. Explaining medical disputes in Chinese public hospitals: the doctor-patient relationship and its implications for health policy reforms. Health Econ Policy Law 2016, 11, 359–378.) is the following: “Overall, the remarkable rise of patients’ consumerist attitudes towards healthcare and the resultant higher expectations, the erosion of social trust in the medical profession, and sensational media coverage have all given rise to various forms of doctor-patient conflict, especially complaints and litigation (Kaba and Sooriakumaran, 2007).” Which means they cited a different source for their claims.
- Line 74: “Such an impact is likely magnified in recent years due to the flourishing development of social media and its power to spread news stories broadly and quickly.” à This sentence needs a source.
- Sources 22 and 23 cannot be accessed.
- Line 88: “recommendations from authoritative journalism” à what is authoritative journalism, what do you mean by that? Do you mean there are authorities who give recommendations for journalists? If so, who are the authorities and what are the recommendations?
- Line 100: “This list is considered an authoritative source” à by whom? Please provide more information. I fear the majority of your readers will not be able to read the Chinese sources you cite. So they need more information.
- Line 120: The previous information in the paper sound as if there were monthly incidents against medical personal, but you report “only” 10 events. Are not all being reported in media outlets?
- Line 124: “We limited searches to news websites with large or moderate readership and impact based on an Alexa ranking greater than 20,000 and a PageRank of 7 or higher.” à What is an Alexa ranking and a PageRank? Please provide more information. Furthermore, why did you use these ranking and not others? How did you arrive at the specific numbers of 20,000 and7?
- Line 153: “irrelevant to the ten selected events” à does that mean you found new events?
- Line 163: “No eligible reports were collected for the other four influential violent events against doctors.” Do you have an idea why there were no reports of these 4 events?
- Line 173: “Following two rounds of extensive group discussions, the team drafted an objective evaluation checklist.” à I appreciate that you invested time and effort into drafting an evaluation checklist. However, I am not clear on how you arrived at the checklist and why you believe the checklist to be objective instead of subjective. Are the items in your checklist ones that appeared in all four considered guidelines?
- Table 1: “a. Use real and complete writer name b. Use real and complete editor name” à Why is it important, that the real names are used? Maybe writer and editor fear repercussions due to their media article and wish to remain anonymous. The information they provide might still be correct and relevant.
- Table 1: “a. Indicate consent from doctors when their perspectives are included b. Indicate consent from patients when their perspectives are included c. Indicate consent from health administrators when their perspectives are included” à Just in case a doctor or hospital administrator really did something wrong and is now unwilling to give consent for this to be published, would a media article about that not still be recommendable?
- Line 183: “Evaluation of the 788 eligible reports was completed by trained raters” à did you check for inter-rater reliability?
- Line 202 and Table 1: “About 40% of the 788 reports inappropriately included details concerning the violent events against doctors (42·1% included the time of the violence, 36·4% the place, 45·4% the methods, and 54·6% the characteristics of the perpetrator(s)).” à It is very confusing that in your table naming the criteria it sounds like media reports should mention the exact details but here it is labelled as inappropriate. I think that it is inappropriate should be made clear from the start. This is corroborated by line 239: “properly present details of the violent events with specific details omitted” à What is a proper representation without details? Maybe you could give an example.
- Line 205 and Table “how to seek help from professional agencies” à It is not quite clear to me, what kind of help is meant. Do you mean help for patients who feel incorrectly treated or help for medical staff who feel in danger, or something else entirely?
- Line 254: Why do you think the government should control the media? Wouldn’t a free press adhering to self-proclaimed ethical guidelines be a better option?
- Line 258: “Inclusion of explicit violence in popular media sources may increase societal interest in violent and dangerous behaviors, arousing interest and possible imitation through social learning and modeling [37,38].” à Please also include and discuss research showing contrary results. One of the sources you cite [17] even states the following: “MYTH Talking about suicide is a bad idea and can be interpreted as encouragement. FACT Given the widespread stigma around suicide, most people who are contemplating suicide do not know who to speak to. Rather than encouraging suicidal behaviour, talking openly can give a person other options or the time to rethink his/her decision, thereby preventing suicide.”
- Line 264: “a goal that may be boosted by offering details about violent events [40]” à I get the feeling that the topics of suicide and violence against others get mixed up a couple of time in this manuscript. The source you cite here is about suicide, but it is used as an argument against detailed media coverage of violence against others.
- Line 266: “It is widely recommended by professional journalism standards that media reports about violence, suicide, drug overdoses, and similar situations include information for readers to seek expert advice for prevention [17,41].” Do these two sources really talk about the aspects you mention in that sentence?
- Line 309: “Policy implications” Sorry, but the whole paragraph sounds highly politically motivated. It is certainly desirable that journalists are educated and that they are educated in all aspects needed to be a good journalist. But ultimately, it is not the government that should regulate the media or even penalise it. Furthermore, to reach such conclusions, you should have reported a direct causal relationship between the media reports you researched and following violence. However, you have not made any points linking your findings to actual violent behaviour. It is all highly speculative.
Minor aspects:
- Line 16: “To examine whether the media reports about violence against doctors in China follow the professional recommendations” – that is not a sentence
- Line 94: “for inclusion” à inclusion in what? At this point in the text I can only guess you are doing a literature search. I think it would be best, if you start your methods section with an introduction of what you did and then go on to be more specific.
- Line 114: was there a specific reason for the chosen years of data inclusion, i.e. 2007-2017?
- Line 260: “appropriately” à approximately
Review summary
I appreciate that you put a lot of effort into the literature search and evaluation. However, for one thing there seems to be a mix-up of suicide and violence against others. And second, I feel your conclusions to be unjustified.
Reviewer 3 Report
The Authors focus on violence against doctors but cite the research related to nurses (line 41- 44). It's difficult to compare those two groups. It should be corrected.
How the Authors define "violence"? What about "aggression"? The Authors use both terms. "Patient and visitors violence" might be helpful.
What was the reason for choosing Alexa ranking as the inclusion criteria?